# Tumor Regression upon Intratumoral and Subcutaneous Dosing of the STING Agonist ALG-031048 in Mouse Efficacy Models

**DOI:** 10.3390/ijms242216274

**Published:** 2023-11-13

**Authors:** Andreas Jekle, Santosh Kumar Thatikonda, Ruchika Jaisinghani, Suping Ren, April Kinkade, Sarah K. Stevens, Antitsa Stoycheva, Vivek K. Rajwanshi, Caroline Williams, Jerome Deval, Sucheta Mukherjee, Qingling Zhang, Sushmita Chanda, David B. Smith, Lawrence M. Blatt, Julian A. Symons, Francois Gonzalvez, Leonid Beigelman

**Affiliations:** 1Aligos Therapeutics, Inc., South San Francisco, CA 94080, USAsstevens@aligos.com (S.K.S.); astoycheva@aligos.com (A.S.); vrajwanshi@aligos.com (V.K.R.); schanda@aligos.com (S.C.); dsmith@aligos.com (D.B.S.); lblatt@aligos.com (L.M.B.); jsymons@aligos.com (J.A.S.); lbeigelman@aligos.com (L.B.); 2Aligos Belgium BV, 3001 Leuven, Belgium

**Keywords:** immuno-oncology, STING agonist, immune checkpoint inhibitor, syngeneic mouse model

## Abstract

Stimulator of interferon genes (STING) agonists have shown potent anti-tumor efficacy in various mouse tumor models and have the potential to overcome resistance to immune checkpoint inhibitors (ICI) by linking the innate and acquired immune systems. First-generation STING agonists are administered intratumorally; however, a systemic delivery route would greatly expand the clinical use of STING agonists. Biochemical and cell-based experiments, as well as syngeneic mouse efficacy models, were used to demonstrate the anti-tumoral activity of ALG-031048, a novel STING agonist. In vitro, ALG-031048 is highly stable in plasma and liver microsomes and is resistant to degradation via phosphodiesterases. The high stability in biological matrices translated to good cellular potency in a HEK 293 STING R232 reporter assay, efficient activation and maturation of primary human dendritic cells and monocytes, as well as long-lasting, antigen-specific anti-tumor activity in up to 90% of animals in the CT26 mouse colon carcinoma model. Significant reductions in tumor growth were observed in two syngeneic mouse tumor models following subcutaneous administration. Combinations of ALG-031048 and ICIs further enhanced the in vivo anti-tumor activity. This initial demonstration of anti-tumor activity after systemic administration of ALG-031048 warrants further investigation, while the combination of systemically administered ALG-031048 with ICIs offers an attractive approach to overcome key limitations of ICIs in the clinic.

## 1. Introduction

The cyclic GMP AMP synthase (cGAS)–stimulator of interferon genes (STING) pathway to activate the innate immune system has been identified as an attractive pharmacologic target in immuno-oncology [1,2]. cGAS is a cytoplasmic protein that produces the second messenger 2′3′ cyclic guanosine monophosphate–adenosine monophosphate (2′3′ cGAMP) upon sensing the presence of double-stranded DNA in the cytosol, originating either from viral or bacterial infection or as a consequence of tumorigenesis (reviewed in [3]). The second messenger, 2′3′ cGAMP, then activates the STING protein located in the membranes of the endoplasmic reticulum and mitochondria, which in turn results in the phosphorylation of the interferon regulatory factor (IRF)-3 via the TANK-binding kinase 1, ultimately leading to the release of type I interferon (IFN) [4]. Activation of the STING signaling cascade in antigen-presenting cells within the tumor microenvironment primes CD8 T-cells to recognize tumor antigens and induces a long-lasting, immunologic anti-tumor response [1,2]. Understanding this molecular pathway triggered the development of STING agonists as cancer immuno-therapeutics [5,6,7,8], with several compounds entering the clinic, such as ADU-S100 (developed by Aduro Biotech/Novartis, NCT03172936); MK-1454 (ulevostinag, Merck Sharp and Dohme Corp., NCT04220866 and NCT03010176); E7766 (Eisai, NCT04144140); GSK3745417 (GlaxoSmithKline, NCT03843359); BMS-986301 (Bristol-Myers Squibb, NCT03956680); TAK-676 (Takeda, NCT04420884); and SB11285 (Spring Bank Pharmaceuticals, NCT04096638) [6,8,9,10,11,12,13,14].

Clinical development of many of the STING agonists focuses on combination therapy with immune checkpoint inhibitors (ICIs) [15,16]. ICIs have been a major breakthrough in cancer therapy but benefit only a subset of patients due to the resistance of tumors to a “cold”, non-immunogenic tumor microenvironment. STING agonists have the potential to induce a “hot”, pro-inflammatory tumor microenvironment and thereby increase the effectiveness of ICI treatment (recently reviewed in [16]). The beneficial interaction of STING agonists with ICIs has been demonstrated in various preclinical cancer models [8,17,18].

A major limitation of first-generation cyclic di-nucleotide (CDN)-type STING agonists is their requirement for intratumoral (IT) administration. CDNs are rapidly degraded in vivo via phosphodiesterases such as ectonucleotide pyrophosphatase/phosphodiesterase (ENPP1) [19]. In addition, the negatively charged, hydrophilic nature of CDNs prevents their diffusion through cell membranes. In this study, we demonstrate how a novel STING agonist, ALG-031048, with greater resistance to phosphodiesterase degradation, can activate primary human dendritic cells ex vivo, induce a strong, long-lasting immunologic anti-tumor response in vivo, and act synergistically with ICIs. Importantly, ALG-031048 showed in vivo anti-tumor efficacy when administered via subcutaneous (SC) injection.

## 2. Results

### 2.1. Increased Stability of ALG-031048

ALG-031048 is a novel 2′3′ CDN STING agonist with a chemical structure consisting of an adenosine monomer with a north-methanocarba sugar modification and a 3′-O methyl-modified guanosine monomer, connected by single phosphorothioate and phosphodiester bonds (Figure 1).

These chemical modifications provide increased in vitro stability in a snake venom phosphodiesterase (SVPD) assay when compared with the natural ligand 2′3′ cGAMP [20] or the clinical-stage CDN ADU-S100 [6,8]. While 2′3′ cGAMP and ADU-S100 were fully degraded after incubation with SVPD for 24 h, ALG-031048 was stable under these conditions (Table 1). Similarly, when the stability was tested using human ectonucleotide pyrophosphatase/phosphodiesterase 1 (ENPP1) [19], which has been identified as the main metabolizing enzyme for CDNs, ALG-031048 had a half-life (t_½_) of >120 min, whereas the t_½_ for 2′3′ cGAMP and ADU-S100 were <30 min and 68 min, respectively (Table 1). Furthermore, ALG-031048 demonstrated high stability in biological matrices such as mouse and human plasma and liver microsomes with t_½_ of >60 and >480 min, respectively (Table 1).

### 2.2. Potent In Vitro Biological Activity of ALG-031048

The potential of ALG-031048 to activate STING was assessed in biochemical and cell-based experiments. First, we measured the ligand binding properties of ALG-031048 relative to 2′3′ cGAMP and ADU-S100 in a biochemical thermal shift binding assay using the most abundant STING isoform, R232. ALG-031048 demonstrated similar binding affinity to STING R232 as the natural ligand 2′3′ cGAMP, with an apparent Kd of 2.71 μM and a maximum temperature shift of 13.8 °C (Figure 2A and Appendix A). ALG-031048 activated the IFN-β and IRF reporter in the HEK 293 STING R232 reporter cell line with half-maximal concentrations of 0.132 and 0.029 μM, respectively (Figure 2B,C and Appendix A).

No reporter activity was observed in HEK 293 cells lacking functional STING, indicating that the cellular stimulation is STING-dependent (Appendix A). ALG-031048 at concentrations up to 1000 nM did not cause any loss of cell viability (Appendix A). Activation of STING was confirmed in the monocytic cell line THP-1, where ALG-031048 induced a dose-dependent release of IP-10 (Appendix A). THP-1 cells naturally express the second most frequent STING isoform, HAQ (R71H-G230A-R293Q), indicating that ALG-031048 can activate multiple STING isoforms.

In addition to established and engineered cell lines, we also assessed the potential of ALG-031048 to activate primary human monocytes and dendritic cells, the main target cells for STING agonists. Monocytes and immature dendritic cells (iDCs) were isolated from human peripheral blood mononuclear cells, incubated with ALG-031048, and analyzed for the maturation and activation markers CD83 and CD86. ALG-031048, 2′3′ cGAMP, and ADU-S100 upregulated the surface expression of CD86 and CD83 after 24 and 72 h, respectively, in both iDCs and to a lesser degree in monocytes (Figure 3, Appendix A). In iDCs, ALG-031048 treatment for 24 h also resulted in the release of key cytokines IFN-β, IP-10, MIP-1α, MCP-1, and TNF-α (Appendix A). At the 24 h time point, either no increase or only a minimal increase was observed for IFN-γ and IL-6.

In summary, the treatment of primary human iDCs and monocytes with the STING agonist ALG-031048 resulted in upregulation of the maturation markers CD83 and CD86 as well as the release of cytokines, including IFN-β.

### 2.3. Potent Tumor Reduction in the CT26 Mouse Tumor Model

Based on improved in vitro stability and the capacity of ALG-031048 to release key cytokines of the STING pathway in cell lines, primary human monocytes, and iDC, we next tested the anti-tumor activity of ALG-031048 in the syngeneic CT26 mouse colon carcinoma model. This model has been shown to be sensitive to STING activators such as ADU-S100, which was included in this study as a positive control [6,21]. Female BALB/c mice bearing subcutaneous CT26 tumors were administered IT doses of 25 or 100 μg of ALG-031048 or ADU-S100 three times 3 days apart (3 × q3d). Dosing started when the tumors reached an average volume of 116 mm^3^. Rapid tumor growth was observed in vehicle-control animals with tumor volumes (TV) of 2000 mm^3^ or more by Day 19 (Figure 4 and Table 2). Treatment with 25 μg of ADU-S100 delayed tumor growth, and tumor regression was observed in 1 of 10 animals by the end of the study (Day 40). Improved anti-tumor activity occurred following the higher dose of ADU-S100 (100 μg) with a dichotomous response: in a subset of mice (44%), complete tumor regression was observed (TV < 10 mm^3^), while in a second subset (44%), tumor growth was delayed compared with vehicle, but the final endpoint of 2000 mm^3^ TV was eventually reached by the end of the study. ALG-031048 at a dose of 25 μg achieved a similar response, with 50% of animals having TV < 10 mm^3^. Remarkably, complete tumor regression (TV < 10 mm^3^) was observed in 90% of animals administered 100 μg of ALG-031048 (Figure 4 and Table 2 and Appendix A). The higher tumor regression rate observed with ALG-031048 coincided with increased cytokines such as IFN-α and -β, TNF-α, interleukin 6 (IL-6), MCP-1, and IP-10 (Appendix A). Either no changes or minimal changes in IL-2, IFN-γ, and IL-12 (p40) were observed.

The strong anti-tumor activity of ALG-031048 was confirmed in the Hepa1–6 mouse hepatocellular carcinoma model, where three IT doses of 100 μg ALG-031048 resulted in a mean tumor reduction of 88%, while a lower dose of 25 μg resulted in a tumor reduction of 12% at the end of the study (Appendix A). In contrast, in vehicle-control animals and in animals administered 10 mg/kg of an anti-PD-1 antibody, 7- and 2-fold increases in tumor volume, respectively, were observed by the end of the study. The reduction in tumor volume after treatment with ALG-031048 was corroborated by a reduction in tumor weight at the end of the study (Appendix A). As with the CT26 mouse tumor model, IT administration of ALG-031048 increased plasma cytokines IFN-β1, IFN-γ, IL-6, MCP-1, MIP-1α, and TNF-α (Appendix A).

### 2.4. Long-Lasting, Antigen-Specific Anti-Tumor Activity

One of the hallmarks of the anti-tumor activity of STING agonists is their potential to induce long-lasting, antigen-specific anti-tumor activity [6,8]. To assess the longevity of the anti-tumor activity of ALG-031048 in the CT26 mouse model, the nine animals that achieved a complete response after three IT doses of 100 μg of ALG-031048 (Figure 4) were re-challenged subcutaneously with CT26 on the opposite flank from the original tumor inoculation. In naïve, age-matched animals, tumors grew rapidly with TV ≥ 2000 mm^3^ by Day 28. In contrast, in 8 of 9 animals that had been administered 100 μg of ALG-031048 3 × q3d in the first part of the study, tumor growth was suppressed for 40 days without additional treatment (Figure 5A). The remaining animals exhibited delayed tumor growth compared with naïve animals, with a TV of 864 mm^3^ at the end of the study.

To address whether the observed long-lasting anti-tumor activity is antigen-specific and therefore immune-mediated, CT26-tumor-bearing animals were administered ALG-031048 as described in Figure 4. In this study, all 10 animals receiving three IT doses of 100 μg of ALG-031048 showed complete tumor regression (TV < 10 mm^3^) by Day 29 and remained tumor-free until Day 60. Tumors in the vehicle-control animals reached a TV of ≥ 2000 mm^3^ by Day 14 (Figure 5B). Next, animals in which the initial CT26 tumor had completely regressed following administration of ALG-031048 were re-challenged with CT26 (left flank) and with antigenically unrelated EMT-6 murine mammary carcinoma cells on the opposite, right flank (Figure 5C,D). Both CT26 and EMT-6 tumors developed rapidly in naïve control animals. In contrast, animals that had been previously administered ALG-031048 in the first part of the study had suppressed growth of the CT26 tumor, with six animals demonstrating complete tumor suppression (TV < 10 mm^3^) and three animals with delayed tumor growth (TV of 40–221 mm^3^ at last measurement) (Figure 5C). However, antigenically unrelated EMT-6 tumors formed in these animals with kinetics comparable to those of naïve animals (Figure 5D). These results indicate that the STING agonist ALG-031048 was able to induce antigen-specific, immune-mediated, long-lasting anti-tumor activity in mice.

### 2.5. Improved Anti-Tumor Activity of ALG-031048 in Combination with ICIs

We employed the CT26 mouse model to understand if co-administration of ALG-031048 with the checkpoint inhibitor anti-CTLA-4 would improve anti-tumor activity compared with treatment with each compound alone. CT26-tumor-bearing mice were administered 5 mg/kg anti-CTLA-4 antibody via intraperitoneal (IP) injection on Day 1, followed by 1 mg/kg on Days 4 and 7. A modest delay in tumor growth occurred compared with vehicle-treated animals (Figure 6). The median time to endpoint (TTE) increased from 17.0 days in vehicle-treated animals to 21.0 days upon anti-CTLA-4 treatment (Table 3 and Appendix A). As shown above, treatment with a suboptimal dose of 25 μg of ALG-031048 resulted in delayed tumor growth (median TTE of 26.0 days). However, no complete response was observed in any of the animals administered single-agent anti-CTLA-4 or ALG-031048 treatment (Figure 6A,B). In contrast, 40% of animals administered a combination of anti-CTLA-4 and ALG-031048 had undetectable tumors at the end of the study (Figure 6C), and the median TTE was 39.5 days (Table 3, Appendix A). These results suggest that a combination of the STING agonist ALG-031048 with the ICI anti-CTLA-4 has a synergistic anti-tumor effect.

### 2.6. In Vivo Anti-Tumor Activity of ALG-031048 upon SC Injection

We demonstrated above that ALG-031048 has strong anti-tumor activity in preclinical models when administered IT. However, IT administration is, in many cases, not feasible in a clinical setting. We therefore tested the potential of SC dosing of ALG-031048 in preclinical models. Dosing CT26-tumor-bearing mice via SC injection with 4 mg/kg ALG-031048 (3 × q3d) delayed tumor growth by 21.0 days compared with the vehicle; a complete response (TV < 10 mm^3^) occurred in one animal in this group at the end of the study (Figure 7B). There was no significant difference in the time to endpoint between ALG-031048 dosed at 4 mg/kg SC and an approximately equivalent amount (100 μg) dosed IT (Appendix A). A lower SC dose of 1 mg/kg ALG-031048 caused a minor delay in tumor growth of 4.0 days (Figure 7A and Table 4). As with the higher dose, there was no significant difference in the time to endpoint between the 1 mg/kg ALG-031048 SC dose and an approximately equivalent dose (25 μg) administered IT (Appendix A). Systemic administration induced a similar plasma cytokine profile as IT administration (Appendix A).

The anti-tumor efficacy of ALG-031048 after SC dosing was confirmed in the MC38 mouse colon carcinoma model expressing human PD-L1 (MC38-hPD-L1). Two SC doses of 0.5 mg/kg ALG-031048 on Days 1 and 8 delayed tumor growth with a reduction of 40.3% in TV by Day 11 (Figure 7D). A combination of ALG-031048-dosed SC with the anti-PDL-1 antibody atezolizumab dosed at 5 mg/kg intraperitoneally further reduced tumor growth, with a mean tumor reduction of 65.9% by Day 11 (Figure 7E and Table 5 and Appendix A). This study confirmed the improved anti-tumor efficacy of the STING agonist ALG-031048 when combined with an ICI such as atezolizumab, as has been shown with the combination with the anti-CTLA-4 antibody (Figure 6). Importantly, it also demonstrated that ALG-031048 has anti-tumor activity when dosed with SC, which is a route of administration with a much wider clinical application. Taken together, the novel STING agonist ALG-031048 has demonstrated anti-tumor activity upon systemic administration in two preclinical tumor models and improved activity when combined with ICIs, suggesting how STING agonists might be used clinically in the future.

## 3. Discussion

The discovery of ICIs has revolutionized cancer therapy and given clinicians a new, powerful treatment option. In the US, antibodies against the programmed death receptor-1 (PD-1), programmed death-ligand 1 (PD-L1), and cytotoxic T-lymphocyte-associated protein 4 (CTLA-4) have been approved by the Food and Drug Administration. ICIs block the interaction of suppressors of the immune system on the surface of T-cells such as PD-1 and CTLA-4 with their respective ligands expressed on tumor or antigen-presenting cells, thereby boosting the activity of anti-tumor T-cells. However, the clinical success of ICIs varies greatly among patients and seems to be dependent on a pro-inflammatory (“hot”) tumor microenvironment (reviewed in [7,16,22,23]. One approach to sensitizing “cold” tumors is by activating the cGAS-STING pathway, which links the innate and adaptive arms of the immune system. Pharmacologic activation of the STING pathway within the tumor triggers upregulation of type I IFNs in antigen-presenting cells, such as dendritic cells. Matured, activated dendritic cells migrate to the draining lymph nodes and cross-prime tumor-specific CD8 T-cells, which can then penetrate the primary tumor as well as distant metastases; in addition, the CD8 T-cells provide a long-lasting immunologic memory [6,11,24]. We have demonstrated here that activating primary human antigen-presenting cells ex vivo with the novel STING agonist ALG-031048 increased the expression of the maturation markers CD83 and CD86 on the cell surface of both dendritic cells and monocytes and released IFN-β and other cytokines (Figure 3). In the mouse CT26 colon carcinoma model, IT administration of ALG-031048 demonstrated highly potent, long-lasting, antigen-specific anti-tumor activity in three principal ways. First, after three doses of 100 μg ALG-031048, tumor growth in all 10 animals was significantly delayed compared with vehicle-treated mice (Figure 4). Importantly, in 9 of 10 animals, tumors regressed and were undetectable at the end of the study (Figure 4). This finding was confirmed in the Hepa1–6 mouse hepatocellular carcinoma model, where ALG-031048 reduced the average tumor volume by 88% (Appendix A). Second, mice treated with ALG-031048 remained tumor-free for approximately 2.5 months after the last dose, the longest the animals were observed in any study, demonstrating the long-lasting nature of the anti-tumor response elicited by the STING agonist ALG-031048. Third, 8 of 9 of the tumor-free animals were resistant to a re-challenge with the same CT26 tumor cells, with a significant delay in tumor growth in the remaining animal (Figure 5). In a repeat experiment, ALG-031048 induced a long-lasting antitumor response, and mice vaccinated with ALG-031048 remained resistant to a re-challenge with CT26 cells but were susceptible to the antigenically unrelated tumor EMT-6, establishing the antigen-specific nature of the anti-tumor response (Figure 6). These experiments, which confirm earlier findings [6], underscore the promise of STING agonists for cancer therapy: a finite treatment resulting in a long-lasting, immune-based anti-tumor effect capable of eliminating the primary tumor as well as metastases (abscopal effect).

In Figure 4, we confirm the in vivo anti-tumor activity of the previously described STING agonist ADU-S100 [6,8]. Remarkedly, ALG-031048 exceeded the anti-tumor activity of ADU-S100, resulting in complete tumor regression in 90% of animals in the CT26 colon carcinoma model and 100% in the Hepa1–6 model. The improved anti-tumor activity of ALG-031048 might be due to its resistance to degradation by nucleases (Table 1), particularly ENPP1, which has been identified as a major metabolizing enzyme for the natural STING ligand 2′3′ cGAMP [19,25]. The resistance to nuclease degradation likely results in a longer t_½_ in vivo, augmenting and prolonging the anti-tumor activity of ALG-031048 in mouse efficacy models. However, while ENPP1 has been identified as the main nuclease-degrading dinucleotide STING agonist [19,25], it should be noted that other degradation pathways might contribute to the overall stability in vivo. This hypothesis could be studied in ENPP1-deficient mice or through therapeutic inhibition of ENPP1. Another contributing factor to the potent overall in vivo activity of ALG-031048 may be its tight binding to the STING protein, as indicated by a low apparent Kd in the thermal shift assay (Figure 2A). The strong binding was confirmed by the potent activation of the IFN-β and IRF-reporters in cell-based assays (Figure 2B,C).

In a Phase I clinical trial, ADU-S100 in combination with the ICI spartalizumab, an anti-PD-1 antibody, demonstrated anti-tumor activity. Of 53 treated patients, 12 showed stable disease, 4 patients had a partial response, and 1 patient had a complete response [14,16]. The limited efficacy of ADU-S100 in this early clinical trial may have been caused by its short terminal t_½_ of 10–23 min [5,14,16]. Given the resistance of ALG-031048 to degradation by nucleases, it may have a prolonged in vivo t_½_ and, therefore, improved anti-tumor activity in the clinic, as has been shown here in the mouse colon carcinoma model. Notably, the combination of ADU-S100 and spartalizumab achieved proof-of-concept anti-tumor activity without dose-limiting toxicity [16], alleviating concerns that a more active and longer-lasting STING agonist might cause unacceptable adverse events in the clinic.

A major limitation of first-generation STING agonists is the requirement for IT administration due to their poor tissue penetration and limited stability. In clinical practice, IT therapy is limited to visible tumors or requires skilled clinicians to treat deep-seated tumors. The heterogeneous architecture of tumors, poorly organized vasculature, increased interstitial fluid pressure, and dense extracellular matrix can further limit the intratumoral distribution of IT-delivered therapeutics [26]. A systemic delivery route would thus increase the clinical application of STING agonists. We therefore tested whether ALG-031048 would provide anti-tumor activity in vivo upon SC administration. In the CT26 colon carcinoma model, a dose-dependent delay in tumor growth was observed. Importantly, at the high dose of 4 mg/kg ALG-031048 SC, one animal was tumor-free at the end of the study, supporting the potential of STING agonists to reverse tumor growth in this aggressive preclinical model. The anti-tumor activity of ALG-031048 after SC dosing was confirmed in the MC38-hPD-L1 mouse model, where a low dose of 0.5 mg/kg caused a significant delay in tumor growth, which was further improved with co-administration of the anti-PD-L1 antibody atezolizumab. This initial demonstration of anti-tumor activity after systemic administration of ALG-031048 warrants further investigation to define dose response, dosing regimens, and combinations with ICIs. At the same time, the combination of systemically administered ALG-031048 with ICIs offers a blueprint for future cancer immunotherapy clinical studies.

The current study is using exclusively subcutaneous tumor models. In contrast to orthotopic models, subcutaneous models allow for an easy, reliable assessment of the tumor volume, intratumoral administration of the study drug, and the inoculation of two tumors on different flanks of the animal. However, there are differences in the tumor microenvironment of orthotopic and subcutaneous tumors, such as tumor-infiltrating immune cells [27]. STING agonists have demonstrated anti-tumoral activity in both subcutaneous and orthotopic models [28,29]. The antitumoral activity of ALG-031048 should therefore be confirmed in an orthotopic tumor model before clinical testing is initiated.

## 4. Materials and Methods

### 4.1. Test Articles

ALG-031048, 2′3′ cGAMP, and ADU-S100 were synthesized at Aligos (Therapeutics, Inc., South San Francisco, CA, USA). The anti-mouse PD-1 antibody clone RMP1–14 was purchased from Bioxcell (Lebanon, NH, USA). The anti-human PD-L1 antibody atezolizumab biosimilar and a matching isotype control were provided by Crownbio.

### 4.2. SVPD and ENPP1 Stability Assay

The hydrolysis activity of STING agonists by ENPP1 was measured in a biochemical assay using previously published methods [19]. Briefly, STING agonists at a concentration of 100 μM were incubated for up to 2 h at 37 °C with 26.5 nM of recombinant human ENPP1 (R&D Systems, Minneapolis, MN, USA) in a buffer containing 50 mM of tris pH 9.5 and 150 mM of NaCl. Reactions were stopped by the addition of ice-cold water, and samples were later heated to 95 °C for 3 min prior to HPLC analysis. SVPD activity was similarly conducted using 0.002 U/μL stock concentration of phosphodiesterase I from *Crotalus adamanteus* venom (Sigma-Aldrich, Burlington, MA, USA) diluted 500× in reaction buffer containing 50 mM of tris pH 8, 5 mM of MgCl_2_, and 100 μM of STING agonist. In this case, the enzymatic reaction was stopped with ~2 volumes of 100 mM of EDTA prior to HPLC analysis.

### 4.3. Stability in Mouse and Human Plasma and Liver Microsomes

The plasma stability assay was carried out on 96-well microtiter plates. The test compounds at 5 µM and the reference compound (propantheline) at 1 µM final concentration were incubated separately at 37 °C with mouse or human plasma for 0, 30, 60, and 240 min. At the end of each incubation time point, 300 μL of the quenching solution (50% acetonitrile, 50% methanol, and 0.05% formic acid) containing the internal standard was added to each well. The incubation plates were sealed, vortexed, and centrifuged at 4 °C for 15 min at 4000 rpm. The supernatant was transferred to fresh plates for LC/MS/MS analysis of the test compounds. The peak area ratios of each test compound over the internal standard were plotted against incubation time, and the t_½_ was calculated from the curve assuming first-order kinetics.

The liver microsomal stability assay was carried out in 96-well microtiter plates. The test compounds at 5 µM and the reference compound (verapamil) at 1 µM final concentration were incubated separately at 37 °C with 0.5 mg/mL liver microsomes, with or without 1 mM NADPH in 100 mM potassium phosphate buffer, pH 7.4 with 3.3 mM MgCl_2_. Each reaction mixture had a volume of 25 µL and a final DMSO concentration of 0.1%. At each of the time points (0, 15, 30, and 60 min), the enzymatic reaction was terminated with the addition of 150 μL of quenching solution (100% acetonitrile, 0.1% formic acid, and the internal standard), and subsequently the mixtures were vortexed vigorously for 20 min and centrifuged at 4000 rpm at 10 °C. The supernatants (80 µL) were transferred to a clean 96-well plate and analyzed via LC/MS/MS. The peak area ratios of each test compound over the internal standard were plotted against incubation time, and the t_½_ was calculated from the curve assuming first-order kinetics.

### 4.4. Thermal Shift Binding Assay

Differential scanning fluorimetry was performed in an Applied Biosystems (Woburn, MA, USA) 7900HT real-time PCR system with a ROX detector set at an excitation and emission of 492 and 610 nm, respectively. Each sample was prepared in a total volume of 40 μL that contained a 5× final concentration of SYPRO orange (Invitrogen, Carlsbad, CA, USA) in buffer (20 mM of HEPES pH 7.5, 150 mM of NaCl, 1 mM of DTT, and 1 mM of MgCl_2_) and 4 μM of STING C-terminal domain protein with and without a test article. All the samples were heated at a rate of 1 °C/min, from 20 °C to 99 °C, at ramp rates of 100% and 1%, respectively, with data collection throughout. The resulting fluorescence intensity from the raw dissociation curve data were used to determine the melting temperature for STING protein alone or with a compound. Melting temperature from protein alone was then subtracted from all melting temperatures of protein in the presence of compound, and a resulting melting temperature vs. compound concentration provided apparent Kd values as generated using a sigmoidal dose–response (variable slope) equation in GraphPadPrism version 8.0 (GraphPad Software, Boston, MA, USA).

### 4.5. HEK 293T R232 Reporter Assay

The 293T-Dual hSTING-R232 cells (Invivogen, San Diego, CA, USA) were plated in 96-well plates at a density of 5 × 10^4^ cells per well in DMEM (Corning, Corning, NY, USA) + 10% fetal bovine serum (FBS) (Sigma, St. Louis, MO, USA), 1% penicillin–streptomycin (Corning), 1% non-essential amino acids (NEAA, Corning), 1% Glutagro™ (Corning), and 1% HEPES (Corning). Assays were set up after allowing cells to adhere for 48 h. Compounds dissolved in water were serially diluted in dosing buffer containing 10 µg/mL of digitonin (MP Biomedicals, Solon, OH, USA). Media were aspirated from the cells, and 50 µL of buffer with compound was added in triplicate. After 30 min at 37 °C, the buffer was aspirated and replaced with 100 µL of supplemented media. Cells were incubated for 20 h at 37 °C with 5% CO_2_. The next day, 20 µL of media was transferred to two new plates, and either 50 µL of QuantiLuc™ (Invivogen) to measure IFN-β expression or 80 µL of QuantiBlue™ (Invivogen) to measure IRF activity were added. The luminescence of plates receiving QuantiLuc™ was measured immediately, while plates with QuantiBlue™ were incubated for 30 min at 37 °C before absorbance at wavelength 620 nm was measured. An aliquot of 100 µL of CellTiter-Glo^®^ (Promega, Madison, WI, USA) was added to the original plate still containing the cells, and luminescence was determined to measure viability. All readouts were measured on an Envision plate reader from Perkin Elmer. Data were analyzed using GraphPad™ Prism’s version 8.0 [Agonist] vs. response—variable slope (four parameters) model.

### 4.6. THP-1 Cytokine Release Assay

THP-1 cells (American Type Culture Collection, Manassas, VA, USA) were maintained in maintenance media consisting of RPMI-1640, 1% Pen-Strep, 1% NEAA, 1% Glutagro™, 1% HEPES (all Corning), and 10% FBS (Sigma-Aldrich). On the day of dosing, cells were pelleted and resuspended in fresh media to a density of 1.6 × 10^6^ cells/mL. An aliquot of 60 µL was added to each well of a 96-well U-bottom plate for a total of 1.0 × 10^5^ cells per well, and the final volume was adjusted to 150 µL using fresh media. Compound dilutions were prepared in maintenance media, and 50 µL of diluted compound were added to wells in duplicate. Cells were incubated with the compound overnight at 37 °C. The next day, plates were centrifuged, and the supernatant was collected in a 96-well flat-bottom plate. Human IP-10 was detected using the ProQuantum Immunoassay kit (Thermo Fisher Scientific, Waltham, MA, USA). Data were analyzed using a standard curve and reported as fold-increase over background. Graphs were generated using GraphPad™ Prism version 8.0.

### 4.7. Activation of Primary Human Monocyte and Dendritic Cells

#### 4.7.1. Isolation of Primary Human Monocytes and Dendritic Cells

Frozen human PBMCs (100 × 10^6^ cells) were obtained from Stem Cell Technologies (Kent, WA, USA). Monocytes were purified from PBMCs using a CD14^+^ pan-monocyte-negative isolation kit and an LS column (Miltenyi Biotec, Bergisch Gladbach, Germany). Magnetically labeled non-target cells were depleted by retaining them within a MACS column in the magnetic field of a MACS Separator, while highly purified unlabeled CD1 D16^−^ cells were collected and used to generate dendritic cells. To obtain immature monocyte-derived dendritic cells (iMDDCs), enriched monocytes were cultured in Mo-DC differentiation medium (Miltenyi Biotec) supplemented with human IL-4 (250 IU/mL, Miltenyi Biotec) and human GM-CSF (800 IU/mL, Miltenyi Biotec) at 37 °C in a T-150 flask in a 5% CO_2_ humidified incubator for 5 days; fresh media was replenished every 2–3 days.

#### 4.7.2. Stimulation of Dendritic Cells

The iMDDCs harvested on Day 5 (25,000 cells/well) after differentiation from normal human CD14+ CD16− monocytes were stimulated with STING agonists for 24–48 h. An aliquot of iMDDCs was also cultured in maturation media for 24–48 h and used as a control containing RPMI 1640, 10% FBS, 2 mM L-glutamine, and human TNF-α (6000 IU/mL, Miltenyi Biotec). The supernatant from the plate was frozen to assess cytokine production, and cells were analyzed via flow cytometry.

#### 4.7.3. Flow Cytometry

Cells were incubated with phosphate-buffered saline (PBS) containing 2% FBS and 2 mM of EDTA (FACS buffer) supplemented with 2 mg/mL of normal human IgG on ice for 15 min to block Fc receptors. The cell suspension was then incubated with a predetermined optimal concentration of the appropriate fluorescent dye-labeled monoclonal antibodies (mAbs) against human cell surface markers on ice for 30 min. The fluorescent dye-labeled mAbs against human cell surface molecules included anti-CD209, anti-CD14, anti-CD11c, anti-CD40, anti-HLA-DR, anti-CD86, and anti-CD83 (BD Biosciences, Franklin Lakes, NJ, USA or BioLegend, San Diego, CA, USA). In addition, a fixable live/dead stain (Thermo Fisher Scientific) at a dilution of 1:1000 was added to exclude dead cells. After several washes with FACS buffer, cells were resuspended in FACS buffer and analyzed on an Attune Nxt flow cytometer (Thermo Fisher Scientific) with FlowJo v10 software (BD Biosciences). An aliquot of the cells to be analyzed served as controls by using “fluorescent minus-one” to establish gates and determine the frequency of positively stained cells. Ultra-comp beads (Thermo Fisher Scientific) were used as single-stained controls to set up compensation on the flow cytometer. The gating strategy is provided in Appendix A.

### 4.8. In Vivo Efficacy Models

#### 4.8.1. Compliance and Animal Welfare

The protocols and any procedures involving the care and use of animals in this study were reviewed and approved by the Institutional Animal Care and Use Committee of Charles River Discovery Sciences (Morrisville, NC, USA) for the CT26 models or CrownBio (Beijing, China) for the Hepa1–6 and MC38-hPD-L1 models, respectively, prior to execution. The care and use of animals were conducted in accordance with the regulations of the Association for Assessment and Accreditation of Laboratory Animal Care. Although this study was not conducted in accordance with the FDA Good Laboratory Practice regulations, 21 CFR Part 58, all experimental data management and reporting procedures were in strict accordance with applicable guidelines and standard operating procedures. Cardboard cylinders and tissue paper were used to enrich the environment. Mice were housed in groups of five per cage to avoid single-housing. Animals were monitored for severe dehydration, hypothermia, abnormal/labored respiration, lethargy, obvious pain, diarrhea, skin lesions, neurological symptoms, impaired mobility (not able to eat or drink) due to significant ascites and an enlarged abdomen, astasia, continuous prone or lateral position, signs of muscular atrophy, paralytic gait, clonic convulsions, tonic convulsions, and persistent bleeding from the body orifice. If any animal presented with clinical issues or if unexpected outcomes were observed, the animal(s) were referred to the attending veterinarian for diagnosis and treatment in consultation with the study director and with the goal of alleviating suffering. Animals were anesthetized with isoflurane via an induction box and maintained via nosecone isoflurane. Throughout the anesthetic period, they were monitored for lack of response to stimuli and appropriate cardiopulmonary function. Animals were sacrificed using high-flow CO_2_ inhalation, and death was assured via cervical dislocation, following the guidance of the American Veterinary Medical Association panel on euthanasia.

#### 4.8.2. Statistical Analysis

An unpaired, nonparametic, two-tailed Mann–Whitney test with a confidence level of 95% was used to assess the statistical significance of different treatment groups (GraphPad™ Prism version 8.3.1). In studies presented in Figure 4, Figure 5A,B, Figure 6 and Figure 7A,B, the number of days from start of treatment to endpoint (tumor volume ≥ 2000 mm^3^) was used to compare groups, while in studies depicted in Figure 5C,D, Figure 7C–E and Appendix A, the tumor volume at the indicated time was used. Statistical analysis is provided in Appendix A. For Appendix A, as well as Appendix A, an unpaired *t*-test was used for statistical analysis.

#### 4.8.3. CT26 Mouse Colon Carcinoma Model

##### Mice

Female BALB/c mice (BALB/cAnNCrl, Charles River Laboratories, Morrisville, NC, USA) were 9 weeks old with a body weight range of 16.8 to 21.9 g on Day 1 of the study. The animals were fed ad libitum water (reverse osmosis, 1 ppm Cl) and NIH 31 Modified and Irradiated Lab Diet^®^, consisting of 18.0% crude protein, 5.0% crude fat, and 5.0% crude fiber. The mice were housed on irradiated Enrich-o’cobs™ Laboratory Animal Bedding in static microisolators on a 12 h light cycle at 20–22 °C (68–72 °F) and 40–60% humidity.

##### Tumor Cell Culture

The CT26 murine colon carcinoma cell line was obtained from the American Type Culture Collection (Manassas, VA, USA) and maintained at Charles River Laboratories Discovery Services (Morrisville, NC, USA) in RPMI-1640 medium containing 10% FBS, 2 mM glutamine, 100 units/mL penicillin G sodium, 100 μg/mL streptomycin sulfate, and 25 μg/mL gentamicin. The cells were cultured in tissue culture flasks in a humidified incubator at 37 °C in an atmosphere of 5% CO_2_ and 95% air.

##### Tumor Implantation and Measurement

Each mouse was inoculated subcutaneously in the right flank with 3 × 10^5^ CT26 cells (in 0.1 mL PBS) for tumor development. Ten days after tumor implantation, designated Day 1 of the study, the animals were sorted into seven groups (*n* = 10/group) with mean tumor volumes of 108 or 116 mm^3^, depending on the study.

##### Treatment, Tumor Growth Measurement, and Analysis

PBS was used as the vehicle control. ALG-031048 and ADU-S100 were dissolved in PBS at concentrations of 2 mg/mL and further diluted if needed. Vehicles and test compounds were administered IT q3d × 3 in a volume of 0.05 mL. SC administration was performed between the shoulder blades at a dosing volume of 10 mL/kg. Tumors were measured twice a week in two dimensions using calipers, and volume was calculated using Formula (1):Tumor Volume (mm^3^) = 0.5 (w^2^ × l)(1)

Here, w = width and l = length, in mm, of a tumor.

Tumor weight was estimated with the assumption that 1 mg is equivalent to 1 mm^3^ of tumor volume.

Animals were euthanized when (1) their tumor reached the endpoint volume of 2000 mm^3^, (2) their body weight loss exceeded 30%, or (3) on the last day of the study, whichever came first. The TTE for analysis was calculated for each mouse by the following Equation (2):TTE = (log_10_ (endpoint volume) − b)/m(2)

TTE is expressed in days, endpoint volume is expressed in mm^3^, b is the intercept, and m is the slope of the line obtained via linear regression of a log-transformed tumor growth data set. Animals with tumors that did not reach the endpoint volume were assigned a TTE value equal to the last day of the study. Animals were weighed on Days 1–5, then twice weekly for the duration of the study. The mice were observed frequently for health and overt signs of any adverse treatment-related effects, and noteworthy clinical observations were recorded.

##### Plasma Cytokine Analysis

Plasma cytokine levels were measured in female BALB/c mice bearing subcutaneous CT26 tumors 4 h after receiving one IT dose of 100 μg of ADU-S100 or ALG-031048, at an average TV of 100 mm^3^. Plasma levels of IFN-α, IFN-β, IL-2, TNF-α, IL-6, IL-12, IP-10, IFN-γ, and MCP-1 (in pg/mL) were measured using the ProcartaPlex 9-Plex (ThermoFisher Scientific) according to the manufacturer’s instructions. Data were processed using Milliplex Analyst software version 5.1 on a MAGPIX instrument (Luminex Corp., Austin, TX, USA).

##### Re-Challenge with CT26

Nine mice that demonstrated complete tumor regression on Day 40 following administration of 100 μg of ALG-031048 were re-challenged via SC injection of 3 × 10^5^ CT26 tumor cells in the left flank, opposite of the original implantation. Ten naïve, age-matched female BALB/c mice were used as controls. The animals did not receive any therapeutic treatment. Tumor growth was measured as described above.

##### Re-Challenge with CT26 and EMT-6

Ten mice that showed complete tumor regression on Day 40 following administration of 100 μg of ALG-031048 were re-challenged via SC injection of 3 × 10^5^ CT26 tumor cells in the left flank, at the same site as the original implantation, as well as with 5 × 10^6^ EMT-6 cells on the right flank. Ten naïve, age-matched female BALB/c mice were used as controls. The animals did not receive any therapeutic treatment. Tumor growth was measured as described above.

#### 4.8.4. Hepa1–6 Mouse Hepatocellular Carcinoma Model

##### Mice

Female C57/BL6 mice (Vital River Laboratories Research Models and Services, Beijing, China) were 6 to 8 weeks old with a body weight range of 17.3 to 21.4 g on Day 1 of the study. The animals were fed ad libitum water (0.2-μm filtered, reverse osmosis, autoclaved), and standard irradiated rodent chow. The mice were housed on autoclaved, crushed corncob bedding, which was changed weekly, on a 12 h light cycle at 20–26 °C and 40–70% humidity.

##### Tumor Cell Culture

The Hepa1–6 tumor cells were maintained in vitro in DMEM supplemented with 10% FBS at 37 °C in an atmosphere of 5% CO_2_ in air. Cells in the exponential growth phase were harvested and quantitated using a cell counter, adjusted to 5 × 10^7^/mL in PBS before tumor inoculation.

##### Tumor Implantation, Dosing, Animal Observation, and Tumor Growth Measurement

Each mouse was inoculated subcutaneously in the right front flank region with Hepa1–6 tumor cells (5 × 10^6^) in 0.1 mL PBS for tumor development. After tumor cell inoculation, the animals were checked daily for morbidity and mortality and for any effects of tumor growth or test articles on behavior such as mobility, food and water consumption, eye/hair matting, and any other abnormalities. Dosing started when the average tumor volume reached 98 mm^3^. The anti-PD-1 antibody was dosed IP at 10 mg/kg twice per week (BIW), while ALG-031048 was dosed IT at 25 and 100 μg/animal on Days 1, 5, and 9. Tumor volumes were measured twice, as described above.

#### 4.8.5. MC38-hPD-L1 Mouse Colon Carcinoma Model

##### Mice

Female C57/BL6 mice (Shanghai Lingchang Biotechnology Co., Ltd., Shanghai, China) were 5 to 8 weeks old with a body weight > 16 g at the time of inoculation. Animals were housed and fed as described above.

##### Tumor Cell Culture and Inoculation

After inoculation with tumor cells, the animals were checked daily for morbidity and mortality. During routine monitoring, the animals were checked for any effects of tumor growth or test articles on behavior such as mobility, food and water consumption, body weight gain/loss (body weights were measured twice per week after randomization), eye/hair matting, and any other abnormalities. Tumor volumes were measured and calculated as described above.

##### Dosing

Each group consisted of 10 animals. The control group was administered 5 mg/kg isotype control antibody in a volume of 10 μL/g of IP, BIW for 2.5 weeks, 5 doses total. The atezolizumab-treated group was administered 5 mg/kg in a volume of 10 μL/g of IP, BIW for 2.5 weeks, 5 doses total. ALG-031048 was administered at 5 mg/kg in a volume of 5 μL/g, SC (between the shoulder blades), weekly for 3 weeks. Dosing started when average tumor volumes were 70 mm^3^.

## 5. Conclusions

ALG-031048 is a second-generation STING agonist with improved stability, in vitro potency, and anti-tumoral activity in mouse syngeneic mouse efficacy models. Systemic administration of ALG-031048 overcomes one of the main limitations of earlier STING agonists. Its synergistic activity with ICIs offers potential new treatment options for cancer patients.

## Figures and Tables

**Figure 1 ijms-24-16274-f001:**
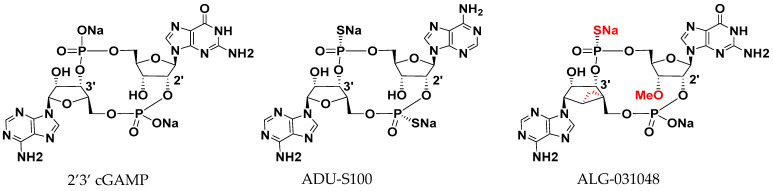
Chemical Structures of 2′3′ cGAMP, ADU-S100, and ALG-031048. Chemical modifications in ALG-031048 relative to 2′3′ cGAMP are highlighted in red.

**Figure 2 ijms-24-16274-f002:**
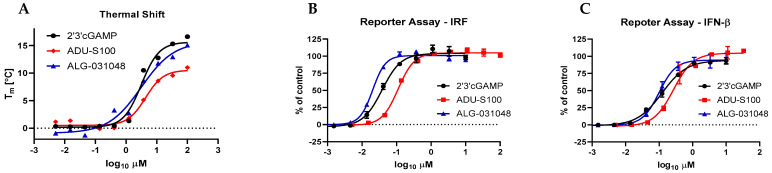
In vitro STING R232 activation of 2′3′ cGAMP, ADU-S100, and ALG-031048 in a biochemical thermal shift assay (**A**), as well as in HEK 293 R232 cells with an IRF (**B**) and IFN-ß reporter (**C**). Shown are the mean of three biological replicates ± SD as error bars of representative dose–response curves.

**Figure 3 ijms-24-16274-f003:**
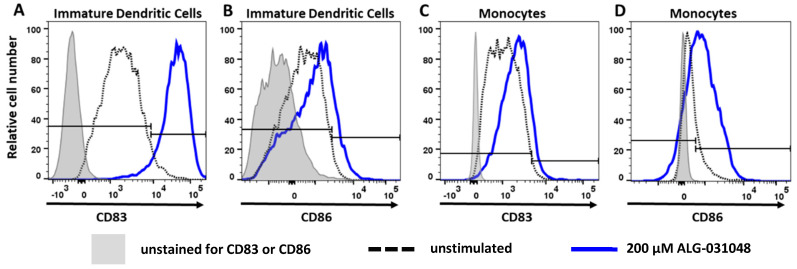
Upregulation of CD83 (**A**,**C**) and CD86 (**B**,**D**) on the surface of primary human immature dendritic cells (**A**,**B**) and monocytes (**C**,**D**). Monocytes were treated for 24 h, while immature dendritic cells were treated for 72 h. Control cells unstained for CD83 and CD86, respectively, shaded in gray, unstimulated cells (dashed black line), and cells stimulated with 200 μM of ALG-031048 (blue line). The gating strategy is provided in Appendix A.

**Figure 4 ijms-24-16274-f004:**
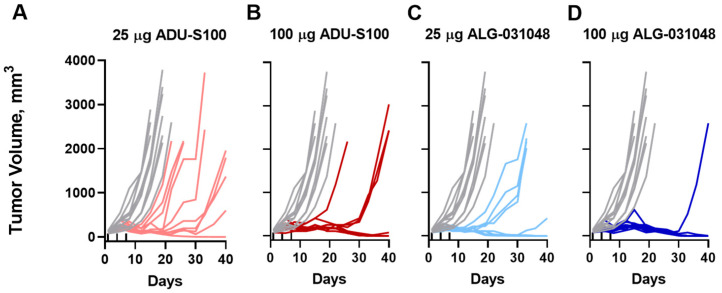
Tumor regression in CT26-tumor-bearing BALB/c mice upon IT treatment of 25 μg (**A**, light red) or 100 μg of ADU-S100 (**B**, dark red) or 25 μg (**C**, light blue) or 100 μg of ALG-031048 (**D**, dark blue) on Days 1, 4, and 7 as indicated with the upward tick marks. Vehicle-treated animals are shown in gray; each line depicts the tumor volume over time of one animal, and there are 10 animals in each group. Treatment started when tumors reached a median volume of 116 mm^3^, 10 days post-implantation.

**Figure 5 ijms-24-16274-f005:**
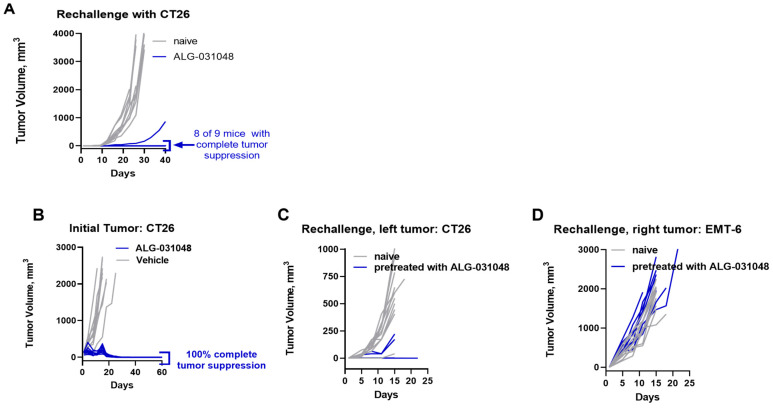
(**A**) ALG-031048 provides long-lasting anti-tumor activity in vivo. Nine CT26-tumor-bearing BALB/c mice, which showed complete tumor regression in Figure 4 after treatment with ALG-031048, were re-challenged with an SC injection of CT26 on the contra-lateral flank. Ten naïve, age-matched mice served as controls. Tumor volume was measured every 3 days. Each line depicts the tumor volume over time in one animal. *p*-value of <0.001 for ALG-031048 pretreated vs. naïve animals based on time to endpoint in days. (**B**) Complete tumor suppression in CT26-bearing BALB/c mice treated with three IT doses of 100 μg of ALG-031048 (blue) compared with vehicle-treated animals (gray). *p*-value of <0.001 for ALG-031048 vs. vehicle based on time to endpoint in days. Treatment started at a median tumor volume of 108 mm^3^, 14 days after implantation. These animals (blue) were next re-challenged with CT26 (**C**) and EMT-6 (**D**), and tumor growth was observed for 25 days without additional therapeutic intervention; naïve animals were used as controls (gray). *p*-value of <0.001 for ALG-031048 vs. vehicle treatment for the CT26 tumors (**C**) and *p*-value of 0.125 (non-significant) for the EMT-6 tumors (**D**), based on tumor volume on the last available measurement.

**Figure 6 ijms-24-16274-f006:**
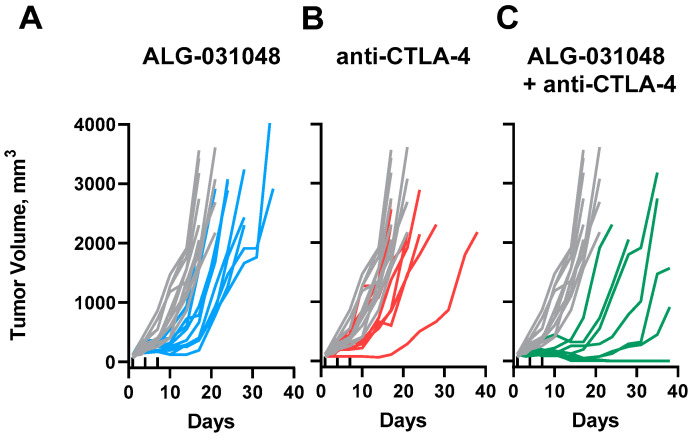
(**A**) Improved anti-tumor activity of CT26-tumor-bearing mice with the combination treatment of anti-CTLA-4 and ALG-031048. CT26 tumor growth curves of animals treated with vehicle (gray) or 25 μg of ALG-031048 dosed intratumorally on Days 1, 4, and 7 (**A**, light blue), or 5 mg/kg anti-CTLA-4 on Day 1, followed by 1 mg/kg anti-CTLA-4 on Days 4 and 7 (**B**, red), or a combination of ALG-031048 and anti-CTLA-4 (**C**, green). Upward tick marks on Days 1, 4, and 7 indicate times of treatment. Treatment started at a median tumor volume of 108 mm^3^, 13 days post-implantation. Tumor volumes were measured every 3 days. Each line depicts the tumor volume over time of one animal, with ten animals in each group.

**Figure 7 ijms-24-16274-f007:**
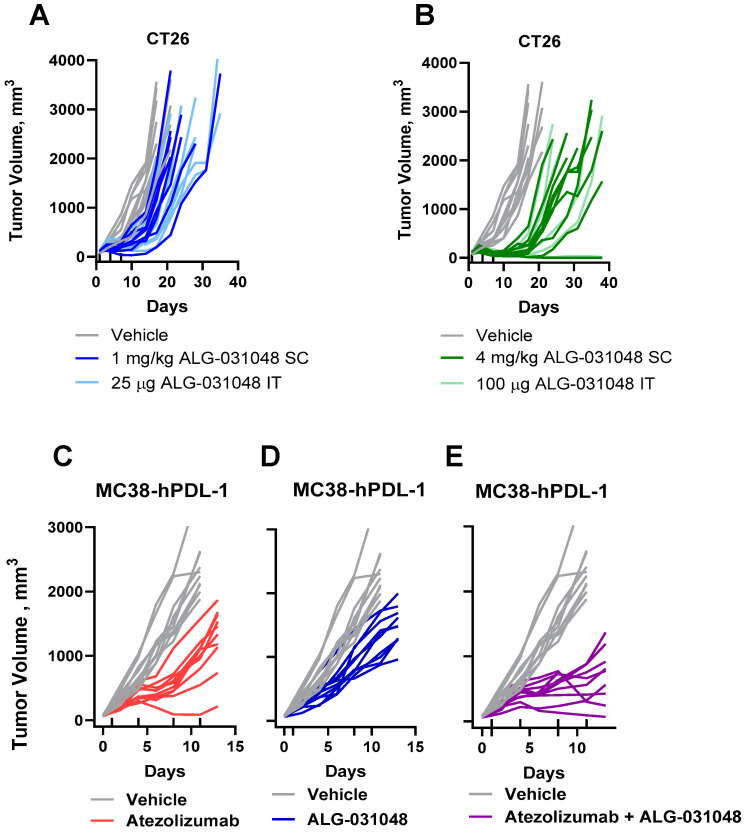
(**A**,**B**): In vivo efficacy after SC dosing of ALG-031048. 10 CT26-tumor-bearing mice per group were treated with SC with 1 (**A**, dark blue) or 4 mg/kg ALG-031048 (**B**, dark green), or IT with 25 μg (**A**, light blue) or 100 μg of ALG-031048. Vehicle-treated control animals are shown in gray. All animals received three treatments on Days 1, 4, and 7, as indicated with the upward tick marks, starting at a mean tumor volume of 108 mm^3^ at 13 days post-implantation. (**C**–**E**) Anti-tumor activity of ALG-031048-dosed SC in the MC38-hPD-L1 mouse tumor model. A total of 10 female MC38-hPD-L1 tumor-bearing C57BL/6 mice per group were treated with 4 BIW doses of 5 mg/kg atezolizumab IP on Days 1, 4, 8, and 11 as indicated with upward tick marks (**C**, red), 2 QW doses of 0.5 mg/kg ALG-031048 SC on Days 1 and 8 as indicated with downward tick marks (**D**, blue), or a combination of ALG-031048 SC and atezolizumab IP (**E**, purple). Vehicle-treated control animals are shown in gray. Treatment was initiated on Day 1 post-implantation, at a median tumor volume of 69 mm^3^.

**Table 1 ijms-24-16274-t001:** Stability of 2′3′ cGAMP, ADU-S100, and ALG-031048 in the SVPD and ENPP1 stability assays and in mouse and human plasma and liver microsomes.

	2′3′ cGAMP	ADU-S100	ALG-031048
**Phosphodiesterase Stability**			
SVPD [% remaining after 24 h]	0	0	100
ENPP1 t_½_ [min]	<30	68	>120
**Plasma t_½_ [min]**			
Mouse	7.2	>60	>60
Human	12.8	>60	>60
**Liver Microsome t_½_ [min]**			
Mouse	ND	>480	>480
Human	ND	>480	>480

ND: not determined; SVPD: snake venom phosphodiesterase; and ENPP1: ectonucleotide pyrophosphatase/phosphodiesterase 1.

**Table 2 ijms-24-16274-t002:** In vivo anti-tumor activity in female BALB/c mice bearing subcutaneous CT26 tumors given three IT doses of vehicle or 25 or 100 μg of ADU-S100 or ALG-031048.

Test Article and Dose (3 × q3d IT, *n* = 10/Group)	Complete Response (TV < 10 mm^3^)	Treatment Failure (TV > 2000 mm^3^)	Delayed Response (TV > 10 and <2000 mm^3^)	Median Time to Endpoint (Days)
Vehicle	0%	100%	0%	19.0
25 μg ADU-S100	10% ^a^	50% ^a^	40% ^a^	36.5
100 μg ADU-S100	44% ^b^	44% ^b^	10% ^b^	40.0
25 μg ALG-031048	50% ^c^	44% ^c^	10% ^c^	40.0
100 μg ALG-031048	90%	10%	0%	40.0

IT = intratumor; 3 × q3d = three times 3 days apart; and TV = tumor volume. ^a^ Four animals with TV ranging from 690 to 1960 mm^3^ at end of study. ^b^ One animal with TV = 75 mm^3^ at end of study; one unrelated death. ^c^ One animal with TV = 405 mm^3^ at end of study. Statistical analysis is provided in Appendix A.

**Table 3 ijms-24-16274-t003:** Improved anti-tumor activity of CT26-tumor-bearing mice with the combination treatment of anti-CTLA-4 and ALG-031048.

Treatment (3 × q3d, *n* = 10/Group)	Complete Response (TV < 10 mm^3^)	Treatment Failure (TV > 2000 mm^3^)	Delayed Response (TV > 10 and <2000 mm^3^)	Median Time to Endpoint (Days)
Vehicle	0%	100%	0%	17.0
25 μg ALG-031048 IT	0%	100%	0%	26.0
5/1 mg/kg anti-CTLA-4 IP	0%	100%	0%	21.0
25 μg ALG-031048 IT +5/1 mg/kg anti-CTLA-4 IP	40%	40%	20% ^a^	39.5

IP = intraperitoneal; 3 × q3d = three times 3 days apart; IT = intratumoral; NA = not applicable; and TV = tumor volume. Note: Tumor growth curves of CT26-bearing BALB/c mice with vehicle or 25 µg of ALG-031048 dosed intratumorally on Days 1, 4, and 7, or 5 mg/kg anti-CTLA-4 on Day 1, followed by 1 mg/kg anti-CTLA-4 on Days 4 and 7 or a combination of ALG-031048 and anti-CTLA-4. ^a^ Two animals with TV of 1568 and 908 mm^3^, respectively, at the end of the study.

**Table 4 ijms-24-16274-t004:** In vivo efficacy after SC dosing of ALG-031048.

Treatment (3 × q3d, *n* = 10/Group)	Complete Response (TV < 10 mm^3^)	Treatment Failure (TV > 2000 mm^3^)	Delayed Response (TV > 10 and <2000 mm^3^)	Median Time to Endpoint (Days)
Vehicle	0%	100%	0%	17.0
1 mg/kg ALG-031048 SC	0%	100%	0%	21.0
25 μg ALG-31048 IT	0%	100%	0%	26.0
4 mg/kg ALG-031048 SC	10%	80%	10% ^a^	35.0
100 μg ALG-31048 IT	40%	50%	10% ^b^	38.0

TV = tumor volume; 3 × q3d = three times 3 days apart; SC = subcutaneous; and IT: intratumoral. Note: Tumor growth curves of CT26-bearing BALB/c mice dosed with vehicle, 25 µg or 100 µg of ALG-031048 intratumorally, or approximately equivalent doses of 1 or 4 mg/kg of ALG-031048 subcutaneously on Days 1, 4, and 7. Dosing started at a mean tumor volume of 108 mm^3^, 13 days post-implantation. ^a^ One animal with TV of 1568 mm^3^ at the end of study; ^b^ one animal with a TV of 18 mm^3^ at the end of study.

**Table 5 ijms-24-16274-t005:** Anti-tumor activity of ALG-031048-dosed SC in the MC38-hPD-L1 mouse tumor model.

Group	Treatment (*n* = 10/Group)	TV (mm^3^) on Day 11 (Median)	*p* Value vs. Group 4	Tumor Reduction vs. Group 1
1	Vehicle	2268	<0.001	NA
2	5 mg/kg Atezolizumab, IP, BIW (4×)	1050	0.017	53.7%
3	0.5 mg/kg ALG-031048, SC, QW (2×)	1353	<0.001	40.3%
4	5 mg/kg Atezolizumab, IP, BIW (4×) + 0.5 mg/kg ALG-031048, SC, QW (2×)	774.0	NA	65.9%

BIW = twice per week; 3 × q3d = three times 3 days apart; IP = intraperitoneal; NA = not applicable; QW = once per week; TV = tumor volume; and SC = subcutaneous. Note: Ten female MC38-hPD-L1-tumor-bearing C57BL/6 mice per group were treated with 4 BIW doses of 5 mg/kg atezolizumab IP, 2 QW doses of 0.5 mg/kg of ALG-031048 SC, or a combination of ALG-031048 SC and atezolizumab IP. Treatment was initiated at an average tumor volume of 70 mm^3^. Tumor volumes on Day 11 are shown.

## Data Availability

Data will be made available upon request.

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
