# Peer review of "Tumor Regression upon Intratumoral and Subcutaneous Dosing of the STING Agonist ALG-031048 in Mouse Efficacy Models"

_ijms, 2023, doi:10.3390/ijms242216274_

Round 1
Reviewer 1 Report
Comments and Suggestions for Authors
The study of Jekle et al. deals with the important problem of immunologically cold tumors and sets to characterize a new STING agonist capable of inducing immune cell penetration and subsequent anti-tumor response. The authors first show how their chemical (ALG-031048) has an improved stability in vitro, while maintaining a very good biological activity. Finally, the extremely encouraging in vivo results on mouse models of colorectal and hepatic cancer testify the feasibility of this approach.
These data should be made available to the scientific community, so I support publication. I only have a few comments that can be addressed in the text:
· The authors should comment on the limits of using a subcutaneous tumor model instead of an orthotopic one in a study that focuses on the tumor microenvironment.
· Why, starting from Table 3, is the column “delayed response” removed? I have the (probably wrong) impression that 25 μg ALG-031048 administered intratumorally was inducing complete response in 10% of mice and delaying tumor progression in 40% of cases (colorectal cancer model, table 2), and then the same treatment did not induce any type of response in another set of 10 mice (table 3). Can the authors comment on this?
· What is the rationale of only using female mice?
· In Fig. 3 it is not specified at which concentration was the drug used
Author Response
We would like to thank reviewer 1 for reviewing the manuscript and their encouraging words.
The comments raised by reviewer 1 are addressed point-by-point below:
Comment 1: The authors should comment on the limits of using a subcutaneous tumor model instead of an orthotopic one in a study that focuses on the tumor microenvironment.
Reply:
We acknowledge the limitation of subcutaneous tumor models and added the following paragraph to the discussion (lines 388-395):
The current study is using exclusively subcutaneous tumor models. In contrast to orthotopic models, subcutaneous models allow for an easy, reliable assessment of the tumor volume, intra-tumoral administration of the study drug and the inoculation of two tumors on different flanks of the animal. However, there are differences in the tumor microenvironment of orthotopic and subcutaneous tumors, such as tumor-infiltrating immune cells [27]. STING agonists have demonstrated anti-tumoral activity in both subcutaneous and orthotopic models [28, 29]. The antitumoral activity of ALG-031048 should therefore be confirmed in an orthotopic tumor model before clinical testing is initiated.
Comment 2: Why, starting from Table 3, is the column “delayed response” removed ? I have the (probably wrong) impression that 25 μg ALG-031048 administered intra-tumorally was inducing complete response in 10% of mice and delaying tumor progression in 40% of cases (colorectal cancer model, table 2), and then the same treatment did not induce any type of response in another set of 10 mice (table 3). Can the authors comment on this?
Reply:
A column “delayed tumor response” has been added to Tables 3 and 4.
The reviewer is correct that in the study depicted in Figure 4/Table 2, 25 μg ALG-031048 IT resulted in a better response than in the studies shown in Figure 6 /Table 3, and Figure 7/Table 4. In all cases, a suboptimal response was observed. The better response seen in Figure 4 can partially be explained with a slower tumor growth in this study in general, as can be observed by the longer median time to endpoint of 19 days in the vehicle group in Figure 4 vs 17.0 days in the studies in Figures 6 and 7. This appears to be variability typically seen in in vivo studies, especially when sub-optimal doses are used.
Comment 3: What is the rationale of only using female mice?
Reply:
The models were initially established and validated with female mice. To reduce variability in terms of body weight and tumor growth, all studies were performed with female animals.
Comment 4: In Fig. 3 it is not specified at which concentration was the drug used.
Reply:
The ALG-031048 concentration shown in Figure 3 was 200 μM. This has been added under the Figure and in the legend. A full dose-response is also provided in supplemental Figure S2.
Reviewer 2 Report
Comments and Suggestions for Authors
In this manuscript, the authors demonstrated the increased in vitro stability, bioactivity, and antitumor activity of the second-generation STING agonist ALG-031048 in a variety of mouse tumor models. Moreover, the authors confirmed the antitumor potency of ALG-031048 administered via subcutaneous injection in two syngeneic mouse tumor models. However, there are several points in this manuscript that require clarification and improvement to enhance its quality and coherency.
1. The majority of experiments in the manuscript still focus on the intratumoral injection of ALG-031048. Therefore, the manuscript title does not closely correspond to the content discussed.
2. The content and format of tables and figures in this manuscript need to be carefully reviewed and adjusted. For example, there is an inconsistency between "tumor volumes" in Table S5 and "tumor weight" in Figure S6B. It is important to ensure consistency in the tables across both the main text and supplementary materials. Additionally, the detection time of CD83 and CD86 is not given specifically in Figures 3 and S2.
3. Table 1 does not provide the specific values of t1/2 for ADU-S100 and ALG-031048 in mouse and human plasma and liver microsomes. Consequently, the stability of the two agonists cannot be compared. Furthermore, it is siginificant to note that the environment in vivo is more complex than that in vitro, and the in vitro experimental data shown in Table 1 may not be sufficient to support the conclusion of "increased stability of ALG-031048."
4. In Table S2, the ED50 of CD83 and CD86 in monocytes after treatment with ALG-031048 is significantly higher than that of 2'3' cGAMP and ADU-S100. This appears to contradict the conclusion that "CD83 and CD86 were upregulated with similar concentrations of ALG-031048, 2'3' cGAMP, and ADU-S100." Please provide some explanations.
5. Figures S4, S5, and S6 do not include an analysis of significant differences for each group of data. Furthermore, in part 2.2, these data presented are not well clarified. Please give more details and explanations.
6. With regards to the experiments conducted on the Hepa1-6 mouse hepatocellular carcinoma model in part 2.3, i) please explain why an anti-PD-1 antibody was chosen instead of ADU-S100 as the control; ii) please provide data or further explanation to support the conclusion that "Necropsy revealed that the animals were, in fact, tumor-free."
7. Please provide data to support the conclusion that "ALG-031048 was well tolerated without any significant body weight loss or adverse clinical observations when administered subcutaneously at 4 mg/kg (3x q3d) to healthy, non-tumor-bearing BALB/c mice" in part 2.6.
8. For part 2.6, ADU-S100 was not chosen as a contrast. It fails to powerfully showcase a significant advantage of ALG-031048 through subcutaneous administration.
Author Response
We would like to thank reviewer 2 for reviewing the manuscript and their insightful comments.
The comments raised by reviewer 2 are addressed point-by-point below:
Comment 1: The majority of experiments in the manuscript still focus on the intratumoral injection of ALG-031048. Therefore, the manuscript title does not closely correspond to the content discussed.
Reply:
We agree. The title was adjusted to “Tumor regression upon intratumoral and subcutaneous dosing of the STING agonist ALG-031048 in mouse efficacy models”.
Comment 2: The content and format of tables and figures in this manuscript need to be carefully reviewed and adjusted. For example, there is an inconsistency between "tumor volumes" in Table S5 and "tumor weight" in Figure S6B . It is important to ensure consistency in the tables across both the main text and supplementary materials. Additionally, the detection time of CD83 and CD86 is not given specifically in Figures 3 and S2.
Reply:
The content and format of tables was carefully reviewed. Figures S5A and S5B now have matching x-axis labels. The incubation times have been added in the legends of Figure 3 and supplemental Figure S2.
Comment 3: Table 1 does not provide the specific values of t1/2 for ADU-S100 and ALG-031048 in mouse and human plasma and liver microsomes. Consequently, the stability of the two agonists cannot be compared. Furthermore, it is significant to note that the environment in vivo is more complex than that in vitro, and the in vitro experimental data shown in Table 1 may not be sufficient to support the conclusion of "increased stability of ALG-031048 ."
Reply:
We agree with the reviewer and addressed this topic in the discussion, lines 345-353:
The improved anti-tumor activity of ALG-031048 might be due to its resistance to degradation by nucleases (Table 1), particularly ENPP1, which has been identified as a major metabolizing enzyme for the natural STING ligand 2’3’ cGAMP [19, 25]. The resistance to nuclease degradation likely results in a longer t1/2 in vivo, augmenting and prolonging the anti-tumor activity of ALG‑031048 in mouse efficacy models. However, while ENPP1 has been identified as the main nuclease degrading dinucleotide STING agonists [19, 25], it should be noted that other degradation pathways might contribute to their overall stability in vivo. This hypothesis could be studied in ENPP1-deficient mice or through therapeutic inhibition of ENPP1.
Comment 4: In Table S2, the ED50 of CD83 and CD86 in monocytes after treatment with ALG-031048 is significantly higher than that of 2'3' cGAMP and ADU-S100. This appears to contradict the conclusion that "CD83 and CD86 were upregulated with similar concentrations of ALG-031048, 2'3' cGAMP, and ADU-S100." Please provide some explanations.
Reply:
We adjusted this statement; it now reads: “ALG-031048, 2’3’ cGAMP, and ADU-S100 upregulated the surface expression of CD86 and CD83 after 24 and 72 hours, respectively, in both iDCs and to a lesser degree in monocytes (Figures 3, S2 and S3, and Table S2)” (lines 117-120).
Comment 5: Figures S4, S5, and S6 do not include an analysis of significant differences for each group of data. Furthermore, in part 2.2 , these data presented are not well clarified. Please give more details and explanations.
Reply:
Cytokine analysis in Figure S4 was performed on single samples. We therefore think that statistical analysis would not be appropriate.
Statistical analysis for Figure S5 was added within the figure (brackets) and in the figure legend.
Statistical analysis for Figures S6a and S6B is provided in Tables S4 and S5. Statistical analysis for Figure 6C has been added to the figure legend.
Additional clarifications have been added to 2.2 (lines 117-120) as well as to the legend of Figure 3.
Comment 6: With regards to the experiments conducted on the Hepa1-6 mouse hepatocellular carcinoma model in part 2.3, i) please explain why an anti-PD-1 antibody was chosen instead of ADU-S100 as the control ; ii) please provide data or further explanation to support the conclusion that "Necropsy revealed that the animals were, in fact, tumor-free. "
Reply:
The aim of the Hepa1-6 study was to confirm the anti-tumoral efficacy of ALG-031048 in a second syngeneic model. The Hep1-6 model has previously been validated with an anti-PD-1 antibody. Since we did not have any experience with a STING agonist in this model, we included the PD-1 antibody as a positive control rather than ADU-S100.
The sentence “Necropsy revealed that the animals were, in fact, tumor-free" was based on observations by the scientist performing the necropsy. Since no photos were taken to further corroborate the finding, this sentence was removed.
Comment 7: Please provide data to support the conclusion that "ALG-031048 was well tolerated without any significant body weight loss or adverse clinical observations when administered subcutaneously at 4 mg/kg (3x q3d) to healthy, non-tumor-bearing BALB/c mice" in part 2.6.
Reply:
While the statement is correct, the data supporting it will be part of a second, upcoming manuscript. We therefore removed this statement from part 2.6 (originally lines 254-256) and 3 (originally lines 378-379).
Comment 8: For part 2.6, ADU-S100 was not chosen as a contrast. It fails to powerfully showcase a significant advantage of ALG-031048 through subcutaneous administration.
Reply:
While ADU-S100 was used as a positive control or comparator in other parts of this manuscript, the aim of the studies in part 2.6 was to test the hypothesis if ALG-031048, on which this manuscript is focused, has antitumoral activity upon subcutaneous administration. We therefore did not include ADU-S100 in these experiments. To our knowledge, in vivo efficacy studies after subcutaneous administration of ADU-S100 have not been performed by the developers of ADU-S100, n
Round 2
Reviewer 2 Report
Comments and Suggestions for Authors
The authors adequately addressed my feedback from the first round of peer review.